# Novel Mitochondrial Gene Rearrangement and Intergenic Regions Exist in the Mitochondrial Genomes from Four Newly Established Families of Praying Mantises (Insecta: Mantodea)

**DOI:** 10.3390/insects13070564

**Published:** 2022-06-21

**Authors:** Yi-Jie Lin, Ling-Na Cai, Yu-Yang Zhao, Hong-Yi Cheng, Kenneth B. Storey, Dan-Na Yu, Jia-Yong Zhang

**Affiliations:** 1College of Chemistry and Life Science, Zhejiang Normal University, Jinhua 321004, China; 945496914@zjnu.edu.cn (Y.-J.L.); cailingna@zjnu.edu.cn (L.-N.C.); superjames@zjnu.edu.cn (Y.-Y.Z.); ydn@zjnu.cn (D.-N.Y.); 2Department of Biology, Carleton University, Ottawa, ON K1S 5B6, Canada; Kenstorey@cunet.carleton.ca; 3Key Lab of Wildlife Biotechnology, Conservation and Utilization of Zhejiang Province, Zhejiang Normal University, Jinhua 321004, China

**Keywords:** Mantodea, gene rearrangement, mitochondrial genome, non-coding region (NCR), phylogenetic relationship

## Abstract

**Simple Summary:**

Mantodea is regarded as an excellent material to study the gene rearrangements and large non-coding regions (LNCRs) in mitochondrial genomes. Meanwhile, as a result of the convergent evolution and parallelism, the gene rearrangements and LNCRs are specific to some taxonomic groups within Mantodea, which play an important role in phylogenetic relationship research. Nine mitochondrial genomes (mitogenomes) from four newly established families of praying mantises are obtained and annotated. Eight types of gene rearrangements, including four novel types of gene rearrangements in Mantodea, are detected, which can be explained by the tandem replication-random loss (TDRL) model. Moreover, one conserved motif between *trnI-trnQ* is detected in Toxoderidae. This study shed light on the formation mechanisms of these gene rearrangements and LNCRs in four newly established families of praying mantises.

**Abstract:**

Long non-coding regions (NCRs) and gene rearrangements are commonly seen in mitochondrial genomes of Mantodea and are primarily focused on three regions: *CR-I-Q-M-ND2*, *COX2-K-D-ATP8,* and *ND3-A-R-N-S-E-F-ND5*. In this study, eight complete and one nearly complete mitochondrial genomes of praying mantises were acquired for the purpose of discussing mitochondrial gene rearrangements and phylogenetic relationships within Mantodea, primarily in the newly established families Haaniidae and Gonypetidae. Except for *Heterochaeta* sp. JZ-2017, novel mitochondrial gene arrangements were detected in *Cheddikulama straminea*, *Sinomiopteryx graham*, *Pseudovates chlorophaea*, *Spilomantis occipitalis*. Of note is the fact that one type of novel arrangement was detected for the first time in the *Cyt b-S2-ND1* region. This could be reliably explained by the tandem replication-random loss (TDRL) model. The long NCR between *trnT* and *trnP* was generally found in Iridopteryginae and was similar to the *ND4L* or *ND6* gene. Combined with gene rearrangements and intergenic regions, the monophyly of Haaniidae was supported, whereas the paraphyly of Gonypetidae was recovered. Furthermore, several synapomorphies unique to some clades were detected that conserved block sequences between *trnI* and *trnQ* and gaps between *trnT* and *trnP* in Toxoderidae and Iridopteryginae, respectively.

## 1. Introduction

Praying mantises (Insecta: Mantodea) are a major group of mimic and predatory insects with over 2400 species in about 29 families and 60 subfamilies, according to the website (http://Mantodea.SpeciesFile.org, accessed on 10 April 2022) [1]. These species are distributed worldwide, mainly in tropical and subtropical areas, and occupy diverse habitats ranging from tropical rainforests to arid deserts [2,3]. There have been numerous studies exploring the taxonomic status within Mantodea based on morphology or molecular data [4,5,6,7,8,9,10,11,12,13,14,15,16,17,18,19]. Existing research confirmed the monophyly of the order Mantodea and Mantodea clustered into Dictyoptera with Blattodea [4,17,20]. However, there remains much bias and controversy surrounding Mantodea, especially over some unstable families or subfamilies, due to the lack of effective molecular datasets. For instance, the monophyly of Tarachodidae and Thespidae, because of the subfamilies Haaniinae and Caliridinae under them, respectively, could not be supported by Zhang et al., [9], Guan et al., [21] or Svenson & Whiting [4,5], whereas they could be supported by Wang et al. [22] under the systematic classification of Ehrmann [2]. It is noteworthy that subtaxa within Mantodea have been changed in the taxonomic system of Schwarz [23] based on the male genital structures. For example, the subfamilies Haaniinae and Caliridinae, as well as genera *Humbertiella* and *Theopompa*, belonged to Thespidae, Tarachodidae, and Liturgusinae (Liturgusidae), respectively, in the past. However, now Haaniidae is comprised of two subfamilies, Haaniinae and Caliridinae, whereas Gonypetidae contains the subfamilies Gonypetinae and Iridopteryginae.

As a class of semi-autonomous organelles, mitochondria are found in most eukaryotic cells [24,25]. In general, the typical mitochondrial genome (mitogenome) of insects is a double-stranded circular structure of 14–20 kb in length and contains 37 genes: 13 protein-coding genes (PCGs), 22 transfer RNA genes (tRNAs), two ribosomal RNA genes (rRNAs), and a large non-coding region with a high A + T content named the control region (CR) or the A + T-rich region [26,27]. Based on the distinguishing characteristics of maternal inheritance, relatively high rates of evolution and variation, rare recombination, and conserved gene components, the mitogenome has proven to be an excellent molecular marker and has been widely utilized in species classification, cryptic species research, and phylogenetic taxonomy [10,12,24,28,29,30,31]. The number of mitogenomes of Mantodea that are available in the NCBI Organelle Genome Resources database (https://www.ncbi.nlm.nih.gov/genome/organelle/, accessed on 21 April 2022) is very high, but still less than 20% of the recorded mantis species. Furthermore, only five complete sequences of Gonypetidae and three complete sequences of Haaniidae were available, so that more mitogenomes from these families are needed in order to understand relationships within Mantodea [8,22,30].

The gene arrangement of almost all insect mitogenomes is compact according to the ancestral phenotype, but exceptions have been detected in some orders, including Hymenoptera, Hemiptera, Ephemeroptera, and Mantodea, among others [28,32,33,34,35,36,37,38]. Gene rearrangements are a relatively common phenomenon in insects, but gene duplications are rare [26]. Gene duplications are mainly concentrated in Mantodea, Hymenoptera, and Hemiptera [6,8,30,39,40,41]. From previous research, particularly in connection with Mantodea, we hypothesized that mantises could be considered as promising candidates for investigating the origin and function of gene rearrangements and gene duplications in mitogenome of insects [6,8,11,30,39]. Three regions of particular interest were detected within the families Gonypetidae, Eremiaphilidae, Deroplatyidae, Hymenopodidae, Toxoderidae, Mantidae: CR-*I*-*Q*-*M*-*ND2*, *COX1*-*K*-*D*-*ATP8,* and *ND3*-*A*-*R*-*N*-*S*-*E*-*F*-*ND5* [6,8,11,30,39]. For example, gene rearrangement in the *K*-*D* cluster was often observed in Toxoderidae mantises such as *Stenotoxodera porioni*, which showed a *COX2-K *-D *-K-D *-K-D *-K-D* cluster (the *K ** or *D ** representing pseudogenes of *K*\*D*) whereas *Toxodera hauseri* showed a *COX2-K *-D *-K-D* cluster [8,11]. Extra *trnR* genes were often detected in the family Hymenopodidae (2–5 tandem duplications), subfamily Mantinae (2–11 tandem duplications), and genus *Theopompa* (2–10 tandem duplications) [8,30]. Meanwhile, extra CR-*I*-*Q*-*M* clusters with several non-coding regions (NCRs) were often found in Mantidae [6,8]. Hence, as a potential evolutionary marker, gene arrangements have been extensively applied in discussing evolutionary patterns and constructing or reconstructing systematic relationships among insect species [26,28,32,36,42,43].

Similarly, both short NCRs and long NCRs (short NCRs ranging from 20 bp to 90 bp and long NCRs over 90 bp) between genes have been widely reported in insects as well as some unique characteristics in several orders or families [6,18,30,35,44,45,46,47,48,49]. For instance, in Ephemeroptera, *trnA*-NCR-*trnR* (with a NCR ranging from 32 to 47 bp) was frequently found in all heptageniid mitogenomes (Ephemeroptera: Heptageniidae) [49]. Within Mantodea, especially Mantidae, NCRs distributed within the CR-*I*-*Q*-*M*-*ND2* region were also frequently detected, which allowed for a huge effort in the reconstruction of Mantidae [6,44]. Short NCRs or long NCRs were commonly considered to be degenerated genes or gene clusters with excessive point mutations and deletions. As a result, they were frequently associated with gene duplication mechanisms [44,50]. The tandem duplication random loss (TDRL) model [50] has been intensively and extensively utilized in explaining the origin and mechanism of gene rearrangements in mitogenomes of mayflies [28,34], bees [32,38], praying mantises [6,8,11,30], and even in crabs [51].

In the present study, we obtained eight complete and one nearly complete mitogenomes from mantises in families: Mantidae Vatinae, Gonypetidae, Thespidae, Haaniidae, and Toxoderidae. Among those, Family Thespidae has not been reported in previous studies due to changes in the taxonomic system [23]. Furthermore, one novel gene arrangement with extra *trnS2* genes was concentrated in the region of *Cyt b-S2-ND1* and several NCRs were distributed in *Spilomantis occipitalis* (*Sp. occipitalis*), so that *Cyt b-S2-ND1* may be the next highlight region in Mantodea, even in insects. Additionally, some gene arrangements were first reported at the family level of Mantodea, such as extra *trnR* genes in Thespidae. This supplementary information could assist in discussing potential gene rearrangement processes and reconstructing phylogenetic relationships of these families in detail.

## 2. Materials and Methods

### 2.1. Sampling Collection

Nine specimens of mantises were collected between 2015 and 2020, including three species of Gonypetidae [*Humbertiella. nada* (*Hu. nada*)*, Theopompa milligratulata*, *Spilomantis occipitalis* (*Sp. occipitalis*)], one Thespidae species (*Carrikerella* sp.), two Haaniidae species [*Haania vitalisi* (*Ha. vitalisi*), *Sinomiopteryx grahami* (*Si. grahami*)], two Toxoderidae species (*Heterochaeta* sp. JZ-2017, *Cheddikulama straminea*), and one Vatinae species (*Pseudovates chlorophaea*). According to the taxonomic system of Schwarz, all specimens were identified based on key diagnostic features of morphology by Dr. J.Y Zhang and with DNA barcoding (Bold Systems v4, http://www.boldsystems.org/, accessed on 23 November 2021) [23]. All samples were stored in 100% ethanol at −40 °C and preserved in the Museum of Zoology, Zhejiang Normal University, China.

### 2.2. DNA Extraction, PCR Amplification and Sequencing

Total genomic DNA was extracted from one leg muscle tissue of a single individual using an Ezup Column Animal Genomic DNA Purification Kit (Sangon Biotech Company, Shanghai, China) and was stored at −20 °C. All mitogenomes were expanded by universal primers with several partial segments [8,52]. As for the vacant part in mitogenomes, several specific primers were designed by Primer Premier 5.0 based on existing sequences [53]. Procedures for normal-PCR and Long-PCR (product length > 3000 bp) are as described in Zhang et al. [8]. Both forward and reverse sequencing of all products were performed using the primer-walking method and AB13730XL by Sangon Biotech Company (Shanghai, China).

### 2.3. Mitogenome Annotation and Analyses

Using DNASTAR Package v.7.1 (Timothy G. Burland, WI, USA), all of the DNA segments were assembled and checked [54]. The nine mitogenomes were then annotated by the MITOS web server (http://mitos.bioinf.uni-leipzig.de/index.py, accessed on 18 February 2022) [55]. Positions and the trefoil secondary structures of tRNAs were confirmed and predicted using the program ARWEN 1.2.3.c (http://130.235.244.92/ARWEN/, accessed on 18 February 2022) and tRNAScan-SE online search Server [56,57]. Additionally, images of tRNA trefoil secondary structure were conducted by a force-directed graph layout (http://rna.tbi.univie.ac.at/forna, accessed on 18 February 2022) [58]. Two rRNAs genes (*16S* and *12S RNA*) and thirteen protein-coding genes (PCGs) were further detected by alignment with the orthologous gene regions of other praying mantis mitogenomes via MEGA 7.0 and Clustal X [59,60]. As for NCRs and CR, NCRs were aligned with flanking regions and results were exported using the website server (https://espript.ibcp.fr/ESPript/ESPript/index.php, accessed on 18 February 2022) [61]. Tandem Repeat Finder V 4.09 (http://tandem.bu.edu/trf/trf.submit.options.ht-ml, accessed on 18 February 2022) was used to search for potential tandem repeat sequences [62]. The CG View Server (http://cgview.ca/, accessed on 19 February 2022) was applied to draw the circular mitogenome maps of the nine mantises [63]. The base components of mitogenomes and relative synonymous codon usage (RSCU) of PCGs were estimated by PhyloSuite [64]. The composition skewness was calculated based on the formulas: AT-skew = (A − T)/(A + T); GC-skew = (G − C)/(G + C) [65].

### 2.4. Phylogenetic Methods

In an attempt to address the phylogenetic relationships with Gonypetidae and Haaniidae, 69 previous mantises mitogenomes, downloaded from NCBI, as well as nine newly combined mitogenomes, were selected as ingroups [6,7,8,10,11,21,22,29,30,39,44,66,67,68,69,70] (Appendix A). Two cockroach species, *Eupolyphaga sinensis* and *Cryptocercus kyebangensis*, and two termites, *Termes hospes* and *Macrotermes barneyi*, were chosen as outgroup taxa [71,72,73]. One dataset (PCG123) was applied for constructing Bayesian inference (BI) and Maximum likelihood (ML) phylogenetic trees that were made up of 13 PCGs with all codon positions. Initial manipulation of the relative data was largely dependent on PhyloSuite [64]. Nucleotide sequences of the 13 PCGs were aligned using MAFFT v 7.475 [74]. Then, conserved regions were confirmed by Gblock 0.91 b and joined the obtained alignments via Geneious 8.1.6 (Matthew Kearse, Auckland, New Zealand) [75,76]. The program PartitionFinder 2.0 (Robert Lanfear, Sydney, Australia) and Bayesian Information Criterion (BIC) was applied to deduce the suitable partitioning strategy and select the best models, respectively [77]. The datasets were defined by each of three codon positions of the 13 PCGs. A total of 14 partitions were found and several best substitution models were displayed in Appendix A. Using RAxML 8.2.0 (Alexandros Stamatakis, Karlsruhe, Germany), ML analysis with the GTRGAMMAI model was run [78]. Moreover, 1000 replicate bootstraps were used to assess the probabilities of branch support (BS). Based on the best estimated model and partition scheme, BI analysis was carried out by using MrBayes 3.2 (Fredrik Ronquist, Stockholm, Sweden) (Appendix A). The BI analysis was simultaneously performed with four chains (one cold chain and three hot chains) for 10 million generations with sampling every 1000 generations [79]. The first 25% of generations were discarded as burn-in. When the average standard deviation of split frequencies was lower than 0.01, BI analysis was judged to have met sufficient convergence.

## 3. Result

### 3.1. Mitogenome Features of Newly Sequenced Mantises

Nine complete or nearly complete mitogenomes were assembled in this study, the *Ha. vitalisi* mitogenome lacking the CR-*I*-*Q*-*M*-partial *ND2* region (Appendix A). The length of the mitogenomes ranged from 13,628 bp in *Ha. vitalisi* to 16,253 bp in *P. chlorophaea*. Only *Heterochaeta* sp. JZ-2017 maintained the typical mitogenome with 13 PCGs, 22 tRNAs, two rRNAs, and 1 CR. On the contrary, tRNA gene rearrangements were detected in all other species (*Hu. nada*, *T. milligratulata*, *Sp. occipitalis*, *Carrikerella* sp., *Si. grahami*, *P. chlorophaea*, *C. straminea*) except *Ha. vitalisi* (Appendix A). Among these mantise mitogenomes, the overlaps and intergenic regions ranged from 19 bp in *C. straminea* to 37 bp in *Si. grahami* and 91 bp in *Heterochaeta* sp. JZ-2017 to 1531 bp in *P. chlorophaea*, respectively. Base compositions were calculated for nine mitogenomes (Table 1). All mitogenomes revealed high A + T content values ranging from 70.1% in *Hu. nada* to 78.6% in *Ha. vitalisi.* Some typical preferences and differences were found in the usage of bases A and C due to a negative GC-skew (−0.292 to −0.151) and positive AT-skew (0.010–0.055) in all mantis mitogenomes. Likewise, rRNAs also featured high A + T content (72.8–80.8%). Contrary to the full sequences, a slightly negative AT-skew (−0.098 to −0.008) and extensively positive GC-skew (0.333–0.470) was seen in rRNAs. Nearly all tRNA genes in these mantises showed the classic cloverleaf secondary structure except for *trnS1* (AGN) in *Si. grahami*, *Sp. occipitalis,* and *Ha. vitalisi* that were missing the dihydrouridine (DHU) arm (Appendix A).

ATN is a classic invertebrate start codon and was widely used in the 12 PCGs, except for *COX1*, which used CTG in *Hu. nada* and *T. milligratulata*), and TTG, which was used in *Sp. occipitalis*, *Ha. vitalisi*, *Heterochaeta* sp. JZ-2017, and *P. chlorophaea*. TAA/T were commonly available stop codons among these mitogenomes, whereas *ND1* in *P. chlorophaea* and *Heterochaeta* sp. JZ-2017, *Cyt b* in *T. milligratulata* and *Si. grahami*, and *ND4* in *Carrikerella* sp. stopped with TAG. *COX3* in *Ha. vitalisi* and *T. milligratulata* stopped with TA. The RSCU of newly sequenced mitogenomes showed that the most popular employed codons (>228) were UUU (F), UUA (L1), and AUU (I), whereas the least used (<7) were UCG (S2), CGC (R), and CCG (P). There was a remarkable similarity in the major codons across these newly sequenced mitogenomes (Appendix A).

### 3.2. Gene Rearrangements

Among nine newly sequenced mantis mitogenomes, four regions of gene arrangement were detected CR-*I*-*Q*-*M*-*ND2*, *COX1*-*K*-*D*-*ATP8*, *ND3*-*A*-*R*-*N*-*S*-*E*-*F*-*ND5*, and *Cyt b-S2-ND1* (Figure 1). Only the gene order pattern of *Heterochaeta* sp. JZ-2017 exhibited the ancestral gene arrangement. In the first region, *Q-M *-I *-Q-M *-I-M-ND2* in *P. chlorophaea*, *CR-M-I-Q-M ** in *Hu. nada*, and *M-CR**′-I-Q-M ** in *T. milligratulata* were discovered (genes marked an extra asterisk mean pseudogenes and CR′ mean partial CR). As for extra *trnM ** and *trnI **, however, possibly due to random loss and transversion, several differences were found compared to the original genes. The anticodon (CAU) of *trnM ** in *P. chlorophaea* turned into CUA (Figure 2). Interestingly, synonymous mutation of an anticodon was found in *trnM ** in *T. milligratulata* from CAU to UAU. The *trnI ** in *P. chlorophaea* was a 35 bp *trnI* residue that lacked the TψC arm and loop and also the partial amino acid arm and anticodon arm (Figure 2). In the second region, *COX2-COX2 *-D-K-D ** was detected in *C. straminea* and the anticodon of the extra *trnD ** gene was different (AUU) (Figure 2). Moreover, a 50 bp sequence named *COX2 ** was found between *COX2* and *trnD* and had a high sequence similarity (72%) to *COX2* in *C. straminea* (Appendix A). In the third region, an extra *trnR* was commonly found in *Carrikerella* sp., *T. milligratulata*, and *Si. graham*, which displayed *A-R-R-R-N*, *A-R-R-R-R-R-R-N*, and *A-R-A*-R*, respectively. The duplicated *trnR* was essentially identical to the original except for genes in *Carrikerella* sp., where the first duplication was consistent with the second but not with the third. Moreover, two identical 19 bp regions were between three *trnR* that suggested degenerated residues of the third. The second *trnA* in *Si. graham* lacked DHU arm and loop and also partial amino acid arm (Figure 2). Additionally, this gene used UAA as a new anticodon, which was presumed to be a degenerated gene. In the last region, surprisingly, *Cyt b-S2-Cyt b *-S2* was detected in *Sp. occipitalis*, which was first spotted in mantis mitogenomes. The 213 bp region between two identical *trnS2* was similar to the *Cyt b* (Appendix A). The anticodon of remaining extra genes in these four regions was consistent with the original matching genes. In summary, most of these gene rearrangements were novel discoveries at the family level, especially the duplicated *trnS2*.

### 3.3. Intergenic Regions

Although most insect mitogenomes are compact with only a few short intergenic regions [26], there were several NCRs (>20 bp) in these newly sequenced mantis mitogenomes. In *C. straminea*, three NCRs (22 bp–28 bp) with high AT% were found between *trnI* and *trnQ*, *trnW* and *trnC*, and *trnY,* and *COX1*, respectively, possibly regarded as hairpin structures. One LNCR between *trnM* and *trnI* in *Hu. nada* had high similarity with the CR, which was consistent with a report by Zhang & Ye [44]. Likewise, several short NCRs between duplicated *trnRs* (G2) were found in *T. milligratulata* and were identical to those reported by Zhang & Ye [44] (Figure 3B). In *P. chlorophaea*, two identical LNCRs (134 bp and 135 bp) found between duplicated *I-Q-M* clusters were partially similar to *ND2* (63%) due to many point mutations. A 40 bp and 37 bp length of the spacer between *ND2* and *trnW*, *trnL2* and *COX2* in *Carrikerella* sp. and *Ha. vitalisi* was similar to *ND2* (75%) and *COX2* (63%) (Appendix A), respectively. By contrast, a long gap with 96 bp between *trnT* and *trnP* was discovered in *Sp. occipitalis*. Interestingly, 34 bp of it showed high similarity (77%) with *ND4L*, and the other was similar to *trnT* (78%) that could form a typical trefoil secondary structure with an abnormal anticodon (UAU) (Figure 3C). The usual gap between *trnS2* and *ND1* was easily detected in all mantis mitogenomes from 14 bp to 37 bp. In general, the CR is the longest intergenic region in all insect mitogenomes. Six of these new mantises mitogenomes were acquired by sequencing (Table 1). The length of the CRs ranged from 673 bp in *Sp. occipitalis* to 1221 bp in *C. straminea* and were positioned between *12S rRNA* and *trnI*. CRs showed higher A + T content (73.8–80.6%) except for *T. milligratulata* (65.1%) with negative GC-skew. As for AT-skew, interestingly, the values were slightly positive except for *Carrikerella* sp. and *Hu. nada*. Furthermore, some repetitive sequences were detected in these CRs. *T. milligratulata* showed 6.2 copy numbers of duplicated consensus sequences of 130 bp and *Heterochaeta* sp. JZ-2017 showed 1.9 copy numbers of duplicated consensus sequences of 21 bp (Figure 3D). The AT% of the duplicated sequences in *T. milligratulata* and *Heterochaeta* sp. JZ-2017 was 62.24% and 84.84%, respectively (Appendix A).

### 3.4. Phylogenetic Analyses

Based on the PCG123 dataset, the phylogenetic relationship within Mantodea was deduced from BI and ML analyses. Few differences were found in the BI and ML trees, but the monophyly of the families except for Amelidae was not disrupted. Where differences were found, the clade tended to display a low value. Hence, we merged the topology common to both trees and used the BI tree as the main topology (Figure 4). *Metallyticus* sp. and *Carrikerella* sp., as representative species of Metallyticidae and Thespidae, respectively, were in a fairly basal position, which was sister to the remaining mantis species (*Metallyticus* sp. + (*Carrikerella* sp. + remaining mantises)). Within Gonypetidae, two subfamilies of Gonypetidae (Gonypetinae and Iridopteryginae) were not sister groups, which meant the Gonypetidae was paraphyletic but the monophyly of two subfamilies was supported. In both trees, Gonypetinae was sister to Nanomantoidea (Leptomantellidae, Amorphoscelidae and Nanomantidae), which formed a cluster as follows: (Gonypetinae + (Leptomantellidae + (Amorphoscelidae + (Nanomantidae)))). Iridopteryginae was clustered together with Haaniidae but not in the ML tree (Appendix A). Because the two subfamilies were independently related in Ehrmann [2] and our result, we tended to elevate these two subfamilies to families (Gonypetidae and Iridopterygidae). In addition, *Heterochaeta* sp. JZ-2017 was moved from Toxoderidae to Amelidae. Hence, the monophyly of Toxoderidae was restored to both BI and ML trees. In short, the monophyly of all families, especially Haaniidae, Gonypetidae, and Iridopterygidae, was supported, although the positions of some families were unstable.

Surprisingly, combined with distribution characteristics of gene rearrangements and NCRs, the monophyly of some families were further confirmed (Figure 4). For instance, the *M-I-Q-M ** cluster was concentrated in Gonypetidae, whereas the additional *trnR* genes were also found in Mantinae and Hymenopodidae. Meanwhile, the gene arrangement in the *K-D* cluster was mainly found in Toxoderidae. The NCR between *trnT* and *trnP* has, to date, been detected in all Iridopterygidae species. Furthermore, tandem duplication and degeneration of the *I-Q-M-ND2* clusters were mainly discovered in Vatinae.

## 4. Discussion

### 4.1. Gene Rearrangements and Rearrangement Mechanisms

Gene rearrangements, regarded as the driving force of evolution, were rarely detected in insect mitogenomes but relatively common in Mantodea, according to previous studies. These have been extensively utilized in the analysis of phylogenetic and homologous evolutionary phenomena [6,8,30,42,43,44,80]. Hence, Mantodea, seen as an excellent group to study, has contributed a more sophisticated understanding of the phenomenon of gene rearrangements. Among our newly sequenced mantis mitogenomes, the gene rearrangement regions covered those seen in previous studies such as the *K-D* cluster and the *A-R-N* cluster [8,11,30], as weak as a new rearrangement region (*Cytb-S2-ND1*). Up to now, explanation of gene rearrangements has commonly used the tandem duplication–random loss (TDRL) model in insects, especially in Mantodea [6,32,35,37,81]. Therefore, this model was equally applicable to this study (Figure 1).

First, in the *CR-I-Q-M-ND2* region, as for *P. chlorophaea*, the *I-Q-M-ND2* cluster was tandem repeated once. Then, the deletion or random loss of the first *ND2* and the second *trnQ* happened, followed by the pseudogenization of the first *trnI* and *trnM*. Then, a tandem repeat of *I *-Q-M **-NCR happened (Figure 1A). This rearrangement mechanism was similar to the description by Xu et al. [6]. The possible explanation for *T. milligratulata* and *Hu. nada* was consistent with Zhang & Ye [44] (Figure 1B). Secondly, in the *COX2-K-D* region, based on comparing the original and current gene arrangement in *C. straminea*, the following scenario was deduced: the tandem duplication of *COX2-K-D* occurred, then the first *K-D* cluster and the second cluster were deleted and repeated, respectively, followed by the deletion of the first *trnK* and the pseudogenization of *COX2*′ and second *trnD* (*COX2*′ indicated partial repetition of *COX2*) (Figure 1C). Likewise, a 174 bp region between *COX2* and *trnK* was discovered in Iridopteryginae JZ-2017 with high similarity to *COX2*, suggesting that it was *COX2*′ [8]. Within Eremiaphiloidea, a similar rearrangement was also found in the Amelidae species (*Yersinia mexicana*), which was regarded as the result of parallel evolution because of the effect of the selection environment [6,82]. Thirdly, in the *ND3*-*A-R-N-S-E-F-ND5* region, the arrangement *A-R-A *-R* was first found in Haaniidae. We interpreted this as one tandem duplication of the *A-R* cluster occurring, followed by degeneration of the second *trnA* (Figure 1D). Multiple additional repetitions of *trnR* were detected in *Carrikerella* sp. and *T. milligratulata* (Figure 1B,E). Due to the identical copies of *trnR* in *T. milligratulata*, a putative explanation was given that six repeats of *trnR* occurred without any point mutation or random loss. However, through alignment, the third repetition was inconsistent with the previous *trnR* genes and two NCRs between extra *trnR* genes were identified that might be a residue of *trnR*. Therefore, we argued that three repetitions of *trnR* occurred. Subsequently, the *A-R-R-R-N* cluster was turned into *A-R-*NCR-*R-N* with point mutation and random loss. Then, the duplication of *R*-NCR happened. Up to now, extra *trnR* genes were widely detected in Mantodea, including in the *Theopompa* species, Mantine species, Hymenopodidae species, and Amelidae species, which appears to be caused by convergent evolution and independent repeats [6,8,30,83]. Finally, this study shows a novel gene rearrangement within Mantodea of *Cyt b-S2-Cyt b *-S2-ND1* in the *Cyt b-S2-ND1* region (Figure 1F). Due to the NCR between duplicated *trnS2* that had high similarity with *Cyt b*, we could infer that *Cyt b-S2* was repeated once, then pseudogenization or random loss of second *Cyt b* followed. An additional *trnS2* residue was discovered in Phyllothelyinae species (*Parablepharis kuhlii asiatica* and *Phyllothelys breve*) [8,84]. With a growing number of mitogenomes available, this region was possibly seen as a hotspot in gene rearrangement.

Due to point mutation and random loss in the absence of selective pressure; duplicated tRNA genes with the abnormal anticodons and vestigial secondary structures were seen as pseudogenes [6,30,42,85]. Such pseudogenes were widely found in the praying mantis [11,39]. For those that had synonymous or missense mutations with intact structures, especially *trnM** in *Hu. nada* and *T. milligratulata*, and *trnD ** in *C. straminea*, the transcriptome analysis was needed to determine whether they were functional or not.

### 4.2. Intergenic Regions

CRs with high AT% are the largest NCRs in the insect mitogenomes and are likely to be involved in the duplication and regulation of transcription of mitogenomes and were hard to obtain [27,86]. Tandem duplications of partial sequences in CRs were extensively detected in Gonypetidae [30]. A unit with 130 bp repeated 6.2 times and 21 bp repeated 1.9 times was also found in *T. milligratulata* and *Heterochaeta* sp. JZ-2017, respectively, representing a good example of the TDRL model. In *T. milligratulata*, after seven repeats of the 130 bp unit, a partial fragment was lost in the seventh. Likewise, in *Heterochaeta* sp. JZ-2017, after one repeat of the 21 bp unit, a partial fragment was lost in the second. Meanwhile, short duplications with a high AT% in the CR were commonly found in Toxoderidae species [11]. Longer lengths with a higher number of repetitions are commonly found in Gonypetidae and Iridopterygidae species. Correspondingly, the CR with repeating units was accordingly longer [8,30] (Appendix A).

The AT content in the CR within Mantodea was at a higher level than in most insects, which meant that it was difficult to obtain the sequences [44]. The CR tended to have a somewhat larger length, particularly in Mantidae where multiple CRs were found several times [6]. Hence, it was of the utmost practical value and significance to utilize existing CR sequences to search for possible highly conserved block sequences (CBSs) within Mantodea. Fortunately, alignment of the CR from four families (Gonypetidae, Iridopterygidae, Toxoderidae and Haaniidae) detected several CBSs within these families or subfamilies (Appendix A). As there was only one species in the family Thespidae, it will not be discussed here. CBS sequences of 268 bp and 226 bp were detected in Gonypetidae and Iridopterygidae, respectively. Two CBSs (47 bp and 90 bp) were also found in both Gonypetidae and Iridopterygidae, except for *T. milligratulata*. Additionally, two CBSs (26 bp and 43 bp) were found in Haaniidae. In consideration of the fact that the control regions of only three Haaniidae species were aligned, due to the lack of the CRs of *Haania* sp. JZ-2017 and *Ha. vitalisi*, it would be necessary to expand the sample size in subsequent studies to verify whether this CBS was plausible. Four CBSs (21 bp, 39 bp, 69 bp and 53 bp) were found in Toxoderidae. Apparently, the lengths and amounts of CBSs within subfamilies were higher than within families and between families. These CBSs could be utilized to design specific primers to split the CR into several segments, which would contribute to obtaining the complete CR for species of the same family or even several families.

Similar to previous studies, a 23–24 bp NCR (G1) between *trnI* and *trnQ* was commonly found in Eremiaphiloidea, except for *Heterochaeta* sp. JZ-2017 [8,11] (Figure 3A). Through sequence alignments, a high similarity was found in G1 within Toxoderidae, shown as a novel motif (TTTYCRTTCCARKAAYTTWATTT) (Figure 3A). Hence, a conjecture was raised that this motif was shared by all Toxoderidae species. An NCR between *trnS2* and *ND1* was also identified that is present in nearly all existing mantis mitogenomes. After alignment of these NCRs, one motif (ACTYAW) was detected and seen as the site of transcription termination factor (DmTTF) action [87].

Basically, the data suggest that duplicated genes were gradually degraded under loose selection pressures and formed pseudogenes or NCRs [42,43]. Hence, the NCRs may be derived from an adjacent gene, which can be explained by the tandem duplication-random loss (TDRL) model [50]. In fact, the condition *ND4L-trnT*-NCR (G3)*-trnP-ND6* was found in *Sp. occipitalis*, 34 bp of which exhibited a high level of likeness to *ND4L*, whereas the other (62 bp) exhibited a high level of likeness to *trnT* (Figure 3C). A viable explanation was given by the TDRL model that after one duplication of *ND4L-trnT*, pseudogenization or random loss happened in the second unit [50]. Furthermore, through comparative analysis within Mantodea, the NCR was unique to this family Iridopterygidae, which had a high similarity to adjacent gene blocks [8]. Except for *Amantis nawai*, the formation of NCR in the remaining Iridopterygidae species was likely similar to *Sp. occipitalis*. As for the NCR in *A. nawai*, it was more similar to the *ND6* gene, which was probably formed after one repeat of *trnP-ND6* and the random loss of the second repeat. On this basis, this phenomenon (*trnT*-NCR-*trnP*) could be a potential hotspot for gene rearrangement and gene degeneration in the Iridopterygidae.

### 4.3. Phylogenetic Analyses

In this study, the monophyly of the families within Mantodea was generally in agreement with Schwarz & Roy [23]. However, nuances were found among these. The monophyly of Haaniidae was supported, which was consistent with Wang et al. [22] and the problem of the paraphyly of Haaniidae in previous studies [4,8,9,11,21] was addressed. However, the location of Haaniidae in both BI and ML trees was unstable. The monophyly of Gonypetidae, Iridopterygidae, and Vatinae was strongly supported (Figure 4). This result was in line with Xu et al. [6] and Zhang et al. [8,9] and provides a strong argument to elevate two subfamilies (Gonypetinae and Iridopteryginae) to families (Gonypetidae and Iridopterygidae). In additionally, within Eremiaphiloidea, *Heterochaeta* sp. JZ-2017 was moved from Toxoderidae to Amelidae. After adjustment, the monophyly of Toxoderidae was recovered, which was consistent with Zhang et al. [11]. However, due to the lack of mitogenomes in Eremiaphilidae and Amelidae, a nuance in phylogenetic relationships was discovered. (((*Heterochaeta* sp. JZ-2017 + *Yersinia mexicana*) + Toxoderidae) + *Schizocephala bicornis*) was detected in BI tree, while (((*Heterochaeta* sp. JZ-2017 + *Schizocephala bicornis*) + *Yersinia mexicana*) + Toxoderidae) was detected in the ML tree (Appendix A). More molecular data is required to fully explore the phylogenetic relationships between Eremiaphilidae and Amelidae.

Intriguingly, incorporating the gene rearrangements and NCR distribution into the phylogenetic analysis, the monophyly of some families was further supported (Figure 4). For example, *trnT*-NCR-*trnP* and *M-I-Q-M *-ND2* were commonly found in Iridopterygidae and Gonypetidae, respectively [8,30]. Gene rearrangement and degeneration in the *COX2-K-D-ATP8* region was unique to Eremiaphiloidea, especially in Toxoderidae [8,11]. This approach has also been applied in previous studies to further investigate the phylogenetic relationships within an order [6,8,9,11,32,39,44,81]. As for disputes that existed, these were raised mainly as a result of the relatively scant dataset with too few mitogenomes in the same family. There may be a need to combine mitogenome data with morphological data, introduced nuclear genes, or more mitogenomes to allow a more comprehensive phylogenetic analysis within Mantodea.

## 5. Conclusions

Nine new mantis mitogenomes were sequences and submitted to the NCBI, including the first from Thespidae (*Carrikerella* sp.). Gene rearrangements were commonly detected in these new sequences, except for *Heterochaeta* sp. JZ-2017. Of these arrangements, the following four novel gene rearrangement patterns were found: *COX2-COX2 *-D-K-D ** in *Cheddikulama straminea*, *A-R-A *-R-N-S-E-F* in *Sinomiopteryx graham*, *Q-M *-I *-Q-M *-I-M-ND2* in *Pseudovates chlorophaea*, *Cyt b-S2-Cyt b *-S2-ND1* in *Spilomantis occipitalis*, and *A-R-R-R-N-S-E-F* in *Carrikerella* sp., which was first detected in Thespidae. Noteworthy among these was the first detection of a gene arrangement in the *Cyt b-S2-ND1* region within Mantodea. Additionally, the LNCR between *trnT* and *trnP* was exclusively found in the Iridopterygidae. Via the tandem duplication-random loss (TDRL) model, appropriate explanations were given for the above phenomena. Furthermore, a considerable number of conserved block sequences in the control region were detected in four families (Gonypetidae, Haaniidae Iridopterygidae, and Toxoderidae) that contributed to the sequencing of the control region and the design of specific primers. Both BI and ML trees supported the monophyly of Gonypetidae, Haaniidae, Iridopterygidae, Toxoderidae, and Vatinae, which was generally consistent with the gene rearrangements and NCR dispositions.

## Figures and Tables

**Figure 1 insects-13-00564-f001:**
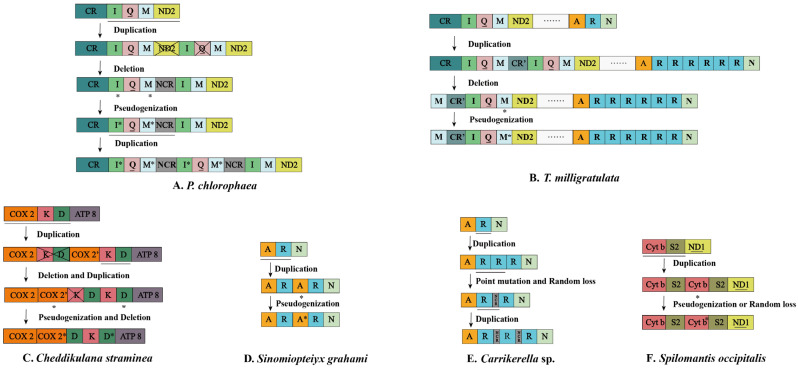
Six mantis mitogenomes with putative mechanism of gene rearrangements. The size of the genes is not proportional. Genes with underlines encoded by the N-strand and those without underlines are genes encoded by the J-strand. Asterisked genes indicate pseudogenes. The remaining genes and gene orders are excluded that were identical to those of the ancestral insect. Horizontal lines, asterisk symbols, and crossed-out colored boxes represent gene duplications, gene pseudogenization, and gene deletions, respectively. (**A**) *Theopompa milligratulata*, (**B**) *Pseudovates chlorophaea*, (**C**) *Cheddikulama straminea*, (**D**) *Carrikerella* sp., (**E**) *Sinomiopteryx grahami*, (**F**) *Spilomantis occipitalis*.

**Figure 2 insects-13-00564-f002:**
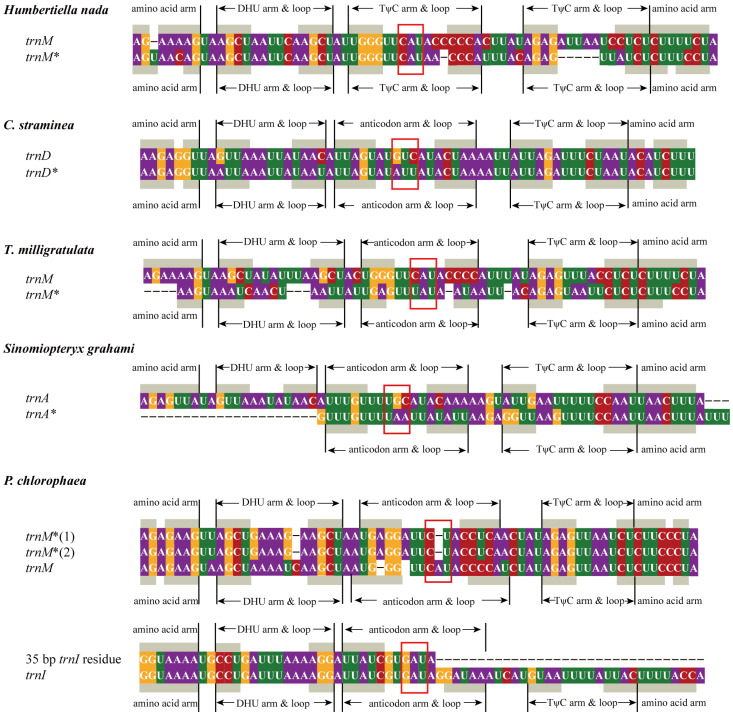
Alignment of *trnM* and pseudogene *trnM* sequences in *Humbertiella nada, T. milligratulata*, and *P. chlorophaea*. Alignment of *trnD* and pseudogene *trnD* sequences in *C. straminea.* Alignment of *trnA* and pseudogene *trnA* sequences in *Sinomiopteryx grahami.* Alignment of *trnI* and 35 bp *trnI* residue in *P. chlorophaea*. Different shaded colors represent different bases. The asterisks (*) denote relevant pseudogene.

**Figure 3 insects-13-00564-f003:**
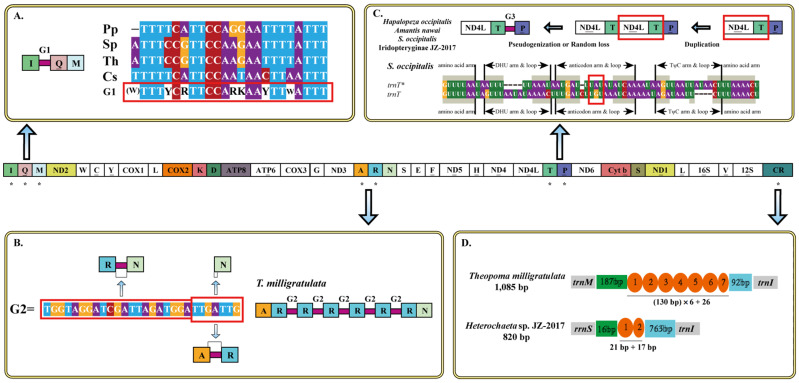
Intergenic gaps are introduced by gene rearrangement and duplication. The gene and genome sizes are not to scale. The majority strand encodes all genes with white blocks, and the minority strand encodes all genes with different colored blocks. The asterisks (*) indicate the gene regions to be displayed and discussed. (**A**) G1 introduced by gene rearrangement in four Toxoderidae mitogenomes. Pp: *Paratoxodera polyacantha*. Sp: *Stenotoxodera porioni*. Th: *Toxodera hauseri*. Cs: *Cheddikulama straminea*. (**B**) G2 introduced by gene duplication in Gonypetidae mitogenomes. (**C**) G3 introduced by gene pseudogenization or random loss in Iridopterygidae. (**D**) Tandem repeats (TDRs) of CRs in *T. milligratulata* and *Heterochaeta* sp. JZ-2017.

**Figure 4 insects-13-00564-f004:**
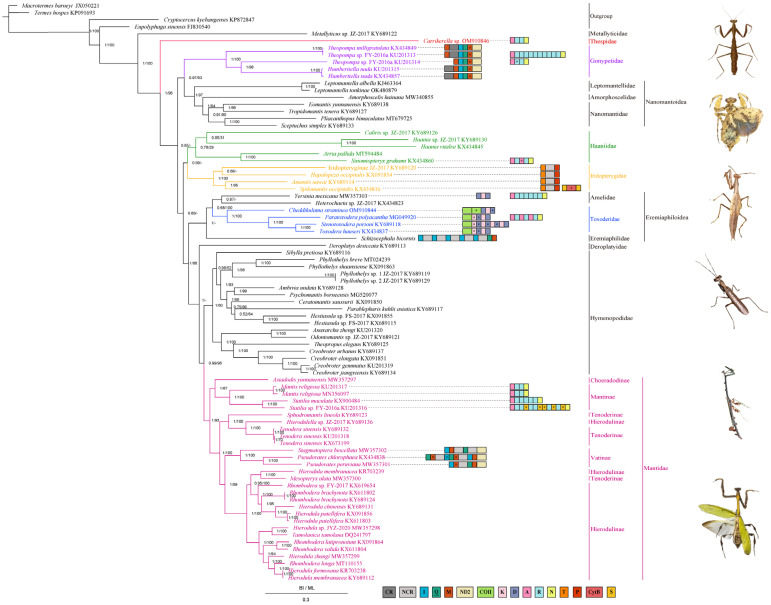
Phylogenetic tree of the relationships among 82 species of Mantodea based on the nucleotide dataset of the 13 mitochondrial protein-coding genes via BI and ML methods. Two cockroaches (*Eupolyphaga sinensis, Cryptocercus kyebangensis*) and two termite (*Termes hospes*, *Macrotermes barneyi*) species was chosen as the outgroup. The numbers on branches display posterior probabilities (PP) and bootstrap support (BS) as determined from BI (left) and ML (right), respectively. The relevant GenBank accession numbers of all species are displayed in the figure. The size of the genes is not proportional. Box images on the right illustrate gene rearrangements and the location of the NCRs for the praying mantis species covered in this study. Genes with underlines encoded by the N-strand and those without underlines are genes encoded by the J-strand. Asterisked genes indicate pseudogenes. The remaining genes and gene orders that were identical to those of the ancestral insect are excluded. The pictures on the far right show the species within the families involved in this study, from top to bottom as followed: Thespidae, Gonypetidae, Haaniidae, Iridopterygidae, Toxoderidae, Vatinae.

**Table 1 insects-13-00564-t001:** Base composition of nine mantis mitogenomes.

Species	A + T (%)				AT-Skew					GC-Skew				
	Mito	PCGs	rRNAs	Contral Region	Mito	PCGs-H	PCGs-L	rRNAs	Contral Region	Mito	PCGs-H	PCGs-L	rRNAs	Contral Region
*Humbertiella nada*	70.1	69.6	72.8	73.8	0.010	−0.109	−0.203	−0.008	−0.031	−0.261	−0.222	0.282	0.414	−0.273
*Theopompa milligratulata*	71.2	71.3	74.5	65.1	0.052	−0.065	−0.224	−0.047	0.085	−0.223	−0.244	0.287	0.371	−0.063
*Spilomantis occipitalis*	76.0	77.1	80.7	80.2	0.055	−0.075	−0.228	−0.098	0.011	−0.151	−0.068	0.224	0.340	−0.233
*Haania vitalisi*	78.6	78.0	80.8	/	0.048	−0.076	−0.216	−0.074	/	−0.220	−0.159	0.296	0.333	/
*Sinomiopteryx grahami*	78.3	78.1	80.3	79.6	0.029	−0.084	−0.204	−0.055	0.031	−0.201	−0.136	0.268	0.393	−0.168
*Pseudovates chlorophaea*	76.8	76.8	79.4	/	0.034	−0.101	−0.239	−0.060	/	−0.169	−0.093	0.288	0.374	/
*Heterochaeta* sp. JZ-2017	75.4	75.3	77.4	80.6	0.026	−0.098	−0.216	−0.028	0.059	−0.253	−0.211	0.327	0.385	−0.228
*Cheddikulama straminea*	75.9	75.9	77.4	77.7	0.035	−0.091	−0.219	−0.069	0.003	−0.212	−0.145	0.300	0.379	−0.184
*Carrikerella* sp.	74.2	73.9	76.1	75.9	0.021	−0.101	−0.196	−0.062	−0.007	−0.292	−0.240	0.350	0.470	−0.260

## Data Availability

The data supporting the findings of this study are openly available from the National Center for Biotechnology Information at https://www.ncbi.nlm.nih.gov, accessed on 21 April 2022. Accession numbers are: OM910844, OM910846 and KX434816, KX434823, KX434838, KX434845, KX434849, KX434857, KX434860.

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
