# Peer review of "Novel Mitochondrial Gene Rearrangement and Intergenic Regions Exist in the Mitochondrial Genomes from Four Newly Established Families of Praying Mantises (Insecta: Mantodea)"

_insects, 2022, doi:10.3390/insects13070564_

Round 1

Reviewer 1 Report

Lin et al generated nine mitogenomes of praying mantises, reconstructed phylogeny based on mitogenomic data, and investigated the evolution of gene rearrangement and intergenic regions. The paper is overall well-written, with a detailed study on gene rearrangement.

The major concern is on the phylogenetic part:

  1. For deep-level relationships, the third codon position is often saturated and therefore not suitable for phylogenetic reconstruction. Many studies either use degenerated nucleotide data, excluding the third codon position for phylogeny or use amino acid data. However, the current study only used one data set, without considering the impact of the third codon position or rRNAs.
  2. For partitioning and model selection, IQTREE might be a handier software for ML tree, which could provide the best partitioning scheme, models, and generate tree in one command line. The current study used PartitionFinder (the latest version PartitionFinder2 was neglected), and all 14 partitions used the GTRGAMMAI model, rather than the best model for each partition.
  3. Insufficient tree searching in ML analysis. Only one tree searching has been done, with 1000 BS replicates. The ideal method should search tree topology multiple times (20-100), select the best topology based on likelihood score, and map 500-1000 BS replicate results on the best topology.

Based on the above concerns, I strongly suggest authors revise the phylogenetic part and perform a more robust analysis.

For other minor comments see the PDF attached.

Author Response

Reviewer 1:

Lin et al generated nine mitogenomes of praying mantises, reconstructed phylogeny based on mitogenomic data, and investigated the evolution of gene rearrangement and intergenic regions. The paper is overall well-written, with a detailed study on gene rearrangement.

Response: Thank you for your comments.

The major concern is on the phylogenetic part:

  1. For deep-level relationships, the third codon position is often saturated and therefore not suitable for phylogenetic reconstruction. Many studies either use degenerated nucleotide data, excluding the third codon position for phylogeny or use amino acid data. However, the current study only used one data set, without considering the impact of the third codon position or rRNAs.

Response: Thank you for your comments.

To begin with the paper, indeed, we used IQTREE 2 [1], MrBayes 3.2 [2] and RAxML 8.2.0 [3] to construct the phylogenetic relationship of Mantodea based on 3 datasets (PCG123: 13 PCGs including all codon positions; PCG12: 13 PCGs without third codon positions; PCGAA: amino acid sequences of 13 PCGs (without MrBayes 3.2)) (Figure. 1-5). Apart from PCGAA, the main topology based on the PCG12 (Figure. 1,2,5), is consistent with the BI tree and ML tree based on the PCG123. By the way, the result of PCGAA was a significant difference from the previous study and taxonomic system (Figure. 4). So, we discard it. The monophyly of Iridopterygidae and Gonypetidae were strongly supported again. By comparing the values of the clades, we find the value of the clades in the phylogenetic relationship based on the PCG123 dataset was higher than the other. Hence, we select it as the final result.

Secondly, via DAMBE 7 [4], we found a slight saturation of third codon positions (Table 1)

Table 1. Saturation testing in phylogenetic analyses.

Gene rigions

NumOTU

Iss

Iss.cSym

P

Iss.cAsym

P

1st condons

32

0.209

0.756

0.0000

0.548

0.0000

2nd condons

32

0.108

0.756

0.0000

0.548

0.0000

3rd condons

32

0.529

0.756

0.0000

0.548

0.0000

All condons

32

0.290

0.781

0.0000

0.573

0.0000

Thirdly, according to previous research [5-12], although there was some saturation of the third codon, these researches used PCG12, PCG123, etc to discuss the phylogenetic relationship within Mantodea. Furthermore, the result of PCG123 was practically consistent with the result of PCG12 or PCG123R. In addition, the value of most clades in the result of PCG123 was higher than the result of PCG12 in Zhang et al [6]. Hence, we think it is fine to use the PCG123 dataset to discuss the phylogenetic relationship within Mantodea in this study. Meanwhile, this study mainly focuses on the structure of mitogenomes of Mantodea. In our opinion, when the third position of codon is slight saturation not high, using all position of codon can obtain the similar phylogenetic relationship with higher bootstrap values or posterior probability values. So, we think our phylogenetic method is no problem.

Furthermore, we used IQTREE 2 and MrBayes 3.2 to construct the phylogenetic relationship of Mantodea based on the PCG123R dataset including all codon positions and two rRNAs (Figure. 6,7). The monophyly of Iridopterygidae and Gonypetidae were strongly supported again. Basically, the main topology is consistent with the BI tree and ML tree based on the PCG123. Definitely, the value of these clades based on the PCG123R dataset was slightly higher than the previous result. However, in this study, the phylogeny within Mantodea is not the most important point. Thank you for reading.

  1. For partitioning and model selection, IQTREE might be a handier software for ML tree, which could provide the best partitioning scheme, models, and generate tree in one command line. The current study used PartitionFinder (the latest version PartitionFinder2 was neglected), and all 14 partitions used the GTRGAMMAI model, rather than the best model for each partition.

Response:Thank you for your comments. We revised the content of the article according to the suggestion. The results obtained by PartitionFinder2 are consistent with the previous one (obtained by PartitionFinder1). All of the 14 partitions belong to the GTR family. And via IQTREE 2 version, we choose the best model for each partition to construct the phylogenetic tree. We find that the results including the main topology and the value of all clades are basically consistent with the previous result (based on the RAxML 8.2.0 method). Thank you for your reading. We did not think IQTREE might be a handier software for ML tree. All analyses methods are neutral.

  1. Insufficient tree searching in ML analysis. Only one tree searching has been done, with 1000 BS replicates. The ideal method should search tree topology multiple times (20-100), select the best topology based on likelihood score, and map 500-1000 BS replicate results on the best topology.

Response:Thank you for your comments. the procedure is followed as “raxmlHPC -x 12345 -p 12345 -# 1000 -m GTRGAMMAI -o Eupolyphaga_sinensis_FJ830540_1, Cryptocercus_kyebangensis_KP872847_1, Termes_hospes_KP091693_1, Macrotermes_barneyi_JX050221_1 -q TL123.txt -s TL123.phy -f a -n TL123ml.tree”. Hence, in fact, we have searched tree topology multiple times.

Based on the above concerns, I strongly suggest authors revise the phylogenetic part and perform a more robust analysis.

Response:Thank you for your suggestions. We analyzed all different methods and found the similar topology in the results. So, we insist on that our methods can resolve phylogenetic problems. Thank you for your suggestions. We can be ready for another paper to compare the different methods or datasets not this paper.

For other minor comments see the PDF attached.

Response: Thank you for your comments. We revised the content of the article according to the suggestion. As followed:

Line 24 Thank you for your comment. We revised it.

Line 27 Thank you for your comment. We revised it.

Line 32 Thank you for your comment. We revised it.

Line 59-61 Thank you for your comment. We revised it.

Line 63 “Since Large non-coding regions is an important part of this study, better to introduce it straight after this sentence.”

Response: Thank you for your comment. In our opinion, for a more logically speaking, it is more suitable to describe the general information about mitochondrion and the mitochondrial genome. Thank you very much. Best wishes.

Line 79 Thank you for your comment. We revised it.

Line 89 “what is the difference between non-coding regions and long non-coding regions”

Response: Thank you for your comment. We are sorry to cause errors in understanding. To begin with, non-coding regions include the control region, and short/long intergenic region in length. We have modified this content. Short NCRs range from 20 bp to 90 bp and long NCRs are generally over 90 bp. We revised all in the paper.

Line 96-97 Thank you for your comment. We revised it.

Line 142 Thank you for your comment. We modified it.

Line 249 Thank you for your comment. We modified it.

Line 252 Thank you for your comment. We have modified it.

Line 371-372 Thank you for your comment. We have modified it.

Line 374 Thank you for your comment. We have modified it.

Line 380 Thank you for your comment. We have modified it.

Some reference in this response.

  1. Minh, B.Q.; Schmidt, H.A.; Chernomor, O.; Schrempf, D.; Woodhams, M.D.; Von Haeseler, A.; Lanfear, R. IQ-TREE 2: new models and efficient methods for phylogenetic inference in the genomic era. Mol. Biol. Evol. 2020, 37, 1530-1534.
  2. Ronquist, F.; Teslenko, M.; Van Der Mark, P.; Ayres, D.L.; Darling, A.; Höhna, S.; Larget, B.; Liu, L.; Suchard, M.A.; Huelsenbeck, J.P. MrBayes 3.2: efficient Bayesian phylogenetic inference and model choice across a large model space. Syst. Biol. 2012, 61, 539-542.
  3. Stamatakis, A. RAxML version 8: a tool for phylogenetic analysis and post-analysis of large phylogenies. Bioinformatics 2014, 30, 1312-1313.
  4. Xia, X.H. DAMBE7: New and Improved Tools for Data Analysis in Molecular Biology and Evolution. Mol. Biol. Evol. 2018, 35, 1550-1552.
  5. Xu, X.D.; Guan, J.Y.; Zhang, Z.Y.; Cao, Y.R.; Storey, K.B.; Yu, D.N.; Zhang, J.Y. Novel tRNA gene rearrangements in the mitochondrial genomes of praying mantises (Mantodea: Mantidae): Translocation, duplication and pseudogenization. Int. J. Biol. Macromol. 2021, 185, 403-411.
  6. Zhang, L.P.; Yu, D.N.; Storey, K.B.; Cheng, H.Y.; Zhang, J.Y. Higher tRNA gene duplication in mitogenomes of praying mantises (Dictyoptera, Mantodea) and the phylogeny within Mantodea. Int. J. Biol. Macromol. 2018, 111, 787-795.
  7. Zhang, L.P.; Yu, D.N.; Cheng, H.Y.; Zhang, J.Y. Data for praying mantis mitochondrial genomes and phylogenetic constructions within Mantodea. Data Brief 2018, 21, 1277-1285.
  8. Zhang, L.P.; Cai, Y.Y.; Yu, D.N.; Storey, K.B.; Zhang, J.Y. Gene characteristics of the complete mitochondrial genomes of Paratoxodera polyacantha and Toxodera hauseri (Mantodea: Toxoderidae). PeerJ 2018, 6, e4595.
  9. Wang, J.-J.; Yang, M.-F.; Dai, R.-H.; Li, H.; Wang, X.-Y. Characterization and phylogenetic implications of the complete mitochondrial genome of Idiocerinae (Hemiptera: Cicadellidae). Int. J. Biol. Macromol. 2018, 120, 2366-2372.
  10. He, B.; Su, T.; Wu, Y.; Xu, J.; Huang, D. Phylogenetic analysis of the mitochondrial genomes in bees (Hymenoptera: Apoidea: Anthophila). PLoS ONE 2018, 13, e0202187.
  11. Bai, J.; Xu, S.; Nie, Z.; Wang, Y.; Zhu, C.; Wang, Y.; Min, W.; Cai, Y.; Zou, J.; Zhou, X. The complete mitochondrial genome of Huananpotamon lichuanense (Decapoda: Brachyura) with phylogenetic implications for freshwater crabs. Gene 2018, 646, 217-226.
  12. Dias, C.; Lima, K.d.A.; Araripe, J.; Aleixo, A.; Vallinoto, M.; Sampaio, I.; Schneider, H.; Rêgo, P.S.d. Mitochondrial introgression obscures phylogenetic relationships among manakins of the genus Lepidothrix (Aves: Pipridae). Mol. Phylogenet. Evol. 2018, 126, 314-320.

Reviewer 2 Report

The manuscript entitled "Novel mitochondrial gene rearrangement and intergenic regions exist in the mitochondrial genomes from four newly established families of praying mantises (Insecta: Mantodea)" has been reviewed and evaluated.
The most merit part of this work is that authors represent novel gene arrangement and intergenic regions in Mantodea mitogenomes. However, previous works (even some of them are published by the authors in the authorship) have showed that gene rearrangement and short or long non-coding regions are common in Mantodea. Therefore, I feel only a few new finding in this work.

The evolutionary processes of these repeat genes or non-coding regions in their Mantodea phylogeny might shed a light on discussing their evolutionary processes. Authors might be focus on this issue and it would be provide lots of new knowledge to readers. Some suggestions are listed as following

1. phylogenetic methods
Some branch supports in your result (Figure 4) are weak, but this may be improved by added tRNA and rRNA genes or removing the thrid codon of PCGs. Authors should try to use different partition schemes (gene or codon partition) to examine the phylogenetic results. In addition, update the phylogenetic softwares. IQ-TREE or RevBayes might be better because they can support different substitution models in different partitions if your phylogenies are based on molecular sequences.

2. the evolution of non-coding regions
Authors have do a lots of effort to hypothesize the evolution of repeat genes and long or short non-coding region. Are they really very useful to be a excellent indicator in grouping (species, genus, subfamily or family level)? Please try to improve this, I think it would be a interesting part to readers.
One of the comparative methods is to mapping these information (the places of duplicate genes or extra non-coding regions) on to the phylogenetic tree. Please have a look on Mesquite project https://www.mesquiteproject.org/.

other comments on this manuscript

In simple summary and Abstract section 
1. the abbreviation of LNCRs, LNCR, NCR, and TDRL. A little redundancy, please concise them.
2. lines 36-37, "novel mitochondrial gene arrangements were detected" and you also present "a novel arrangement was detected for the first time". This writing a little confusing on how many "novel" mitochondrial gene arrangements were found in this manuscript? Please revise this. 

Introduction section
1.line 50, "according to the website [1,2]", but the reference 1 is a article, not a website.
2. The important of gene arrangement, and the arrangement pattern in Mantodea and other insects.
3. the important of non-coding regions, and long non-coding regions.
4. What is the tandem replication-random loss (TDRL) model, you have show in "simple summary" and "Abstract" sections, but no further explaination in "introduction" section.

Materials and Methods section
Phylogenetic methods
1. About the using dataset, authors used protein-coding genes to infer the phylogenetic tree, however, previous studies (Dr. Stephen L. Cameron group, and other research teams) have show that rRNAs and tRNAs also have informative to improve phylogeny. I suggest authors to use different dataset to test the utility of mitogenomic sequences.
2. author should update to PartitionFinder 2. the version 1 is older.
3. different substitution model and partition might obtain different phylogenetic relationships, the author use gene prior to decide partition scheme. How about codon partition? because we know the third position often highly saturation

Results section
Gene rearrangements
1.lines 185-198, please indicate the symbols meaning of first "*" or "'" in the text, I do not like to guess what they are.
2.Figure 2. how decide which one is original transfer gene, and another one is pseudogene?
3. Figure 3. about the box D, the case Theopoma milligratulata show the structure between trnM to trnI. This is not correct region in CR.
4. the resolution is provided figures is not enough, and the word in the figure could not be searched. This is helpful to readers to search certain taxa.
5. the scientific name could not be searched in Figure 4, this is hard to find certain taxa if readers are interested in them.

Discussion
1. lines 284-286, the explaination of TDRL should be place in "introduction" section. This work is address on this part, but it is weird that the explaination is showed in discussion section

Author Response

To reviewer 2:

The manuscript entitled "Novel mitochondrial gene rearrangement and intergenic regions exist in the mitochondrial genomes from four newly established families of praying mantises (Insecta: Mantodea)" has been reviewed and evaluated.
The most merit part of this work is that authors represent novel gene arrangement and intergenic regions in Mantodea mitogenomes. However, previous works (even some of them are published by the authors in the authorship) have showed that gene rearrangement and short or long non-coding regions are common in Mantodea. Therefore, I feel only a few new finding in this work.

Response:Thank you for your comments. Most of the gene rearrangements in this study were detected in the regions where gene rearrangements frequently occurred, but these are first gene rearrangements identified that differ from previous. Notably, one gene rearrangement (Cyt b-S2-Cyt b*-S2) in Cyt b-S2-ND1 was first detected in mitogenomes of Mantodea.

The evolutionary processes of these repeat genes or non-coding regions in their Mantodea phylogeny might shed a light on discussing their evolutionary processes. Authors might be focus on this issue and it would be provided lots of new knowledge to readers. Some suggestions are listed as following

Response:Thank you for your comments. We have modified the article accordingly.

  1. phylogenetic methods
    Some branch supports in your result (Figure 4) are weak, but this may be improved by added tRNA and rRNA genes or removing the thrid codon of PCGs. Authors should try to use different partition schemes (gene or codon partition) to examine the phylogenetic results. In addition, update the phylogenetic softwares IQ-TREE or RevBayes might be better because they can support different substitution models in different partitions if your phylogenies are based on molecular sequences.

Response: Thank you for your comments.

To begin with, indeed, we used IQTREE 2 [1], MrBayes 3.2 [2] and RAxML 8.2.0 [3] to construct the phylogenetic relationship of Mantodea based on 3 datasets (PCG123: 13 PCGs including all codon positions; PCG12: 13 PCGs without third codon positions; PCGAA: amino acid sequences of 13 PCGs (without MrBayes 3.2)) (Figure. 1-5). Apart from PCGAA, the main topology based on the PCG12 (Figure. 1,2,5), is consistent with the BI tree and ML tree based on the PCG123. By the way, the result of PCGAA was a significant difference from the previous study and taxonomic system (Figure. 4). So, we discard it. The monophyly of Iridopterygidae and Gonypetidae were strongly supported again. By comparing the values of the clades, we find the value of the clades in the phylogenetic relationship based on the PCG123 dataset was higher than the other. Hence, we select it as the final result.

Secondly, via DAMBE 7 [4], we found a slight saturation of third codon positions (table 1)

Table 1. Saturation testing in phylogenetic analyses.

Gene rigions

NumOTU

Iss

Iss.cSym

P

Iss.cAsym

P

1st condons

32

0.209

0.756

0.0000

0.548

0.0000

2nd condons

32

0.108

0.756

0.0000

0.548

0.0000

3rd condons

32

0.529

0.756

0.0000

0.548

0.0000

All condons

32

0.290

0.781

0.0000

0.573

0.0000

Thirdly, according to previous research [5-12], Although there was some saturation of the third codon, these researches used PCG12, PCG123, etc to discuss the phylogenetic relationship within Mantodea. Furthermore, the result of PCG123 was practically consistent with the result of PCG12 or PCG123R (13 PCGs including all codon positions and RNA and rRNA). In addition, the value of most clades in the result of PCG123 was higher than the result of PCG12 in Zhang et al [6]. Hence, we think it is fine to use the PCG123 dataset to discuss the phylogenetic relationship within Mantodea in this study. Meanwhile, this study focuses on the structure of mitogenomes of Mantodea.

Furthermore, we used IQTREE 2 and MrBayes 3.2 to construct the phylogenetic relationship of Mantodea based on the PCG123R dataset including all codon positions and two rRNAs (Figure. 6,7). The monophyly of Iridopterygidae and Gonypetidae were strongly supported again. Basically, the main topology is consistent with the BI tree and ML tree based on the PCG123. Definitely, the value of these clades based on the PCG123R dataset was slightly higher than the previous result. However, in this study, the phylogeny within Mantodea is not the most important point. Thank u for reading.

  1. the evolution of non-coding regions
    Authors have done a lot of effort to hypothesize the evolution of repeat genes and long or short non-coding region. Are they really very useful to be an excellent indicator in grouping (species, genus, subfamily or family level)? Please try to improve this, I think it would be an interesting part to readers.
    One of the comparative methods is to mapping these information (the places of duplicate genes or extra non-coding regions) on to the phylogenetic tree. Please have a look on Mesquite project https://www.mesquiteproject.org/.

Response:Thank you for your comments. several explanations and hypothesis were given by TDRL model. The TDRL model was a useful model to explain the gene arrangement, especially in mitogenomes of Mantodea [5,6,8]. Thank you for providing the great method“Are they really very useful to be an excellent indicator in grouping (species, genus, subfamily or family level)?. We find some species shared the similar structure, so we showed those with different color box in the figure 4 of paper. We will learn the Mesquite project https://www.mesquiteproject.org/ in the future. Thanks.

other comments on this manuscript

In simple summary and Abstract section 
1. the abbreviation of LNCRs, LNCR, NCR, and TDRL. A little redundancy, please concise them.

Response:Thank you for your comments. We revised the content of the article according to the suggestion.

  1. lines 36-37, "novel mitochondrial gene arrangements were detected" and you also present "a novel arrangement was detected for the first time". This writing a little confusing on how many "novel" mitochondrial gene arrangements were found in this manuscript? Please revise this. 

Response:Thank you for your comments. We revised the content of the article according to the suggestion.

Introduction section
1.line 50, "according to the website [1,2]", but the reference 1 is a article, not a website.

Response:Thank you for your comments. We revised the content of the article according to the suggestion.

  1. The important of gene arrangement, and the arrangement pattern in Mantodea and other insects.

Response:Thank you for your comments. We revised the content of the article according to the suggestion. Relevant content has been added.

  1. the important of non-coding regions, and long non-coding regions.

Response:Thank you for your comments. We revised the content of the article according to the suggestion. Relevant content has been added.

  1. What is the tandem replication-random loss (TDRL) model, you have show in "simple summary" and "Abstract" sections, but no further explaination in "introduction" section.

Response:Thank you for your comments. We revised the content of the article according to the suggestion. Relevant content has been added.

Materials and Methods section
Phylogenetic methods
1. About the using dataset, authors used protein-coding genes to infer the phylogenetic tree, however, previous studies (Dr. Stephen L. Cameron group, and other research teams) have show that rRNAs and tRNAs also have informative to improve phylogeny. I suggest authors to use different dataset to test the utility of mitogenomic sequences.

Response: To begin with, indeed, we used IQTREE 2 [1], MrBayes 3.2 [2] and RAxML 8.2.0 [3] to construct the phylogenetic relationship of Mantodea based on 3 datasets (PCG123: 13 PCGs including all codon positions; PCG12: 13 PCGs without third codon positions; PCGAA: amino acid sequences of 13 PCGs (without MrBayes 3.2)) (Figure. 1-5). Apart from PCGAA, the main topology based on the PCG12 (Figure. 1,2,5), is consistent with the BI tree and ML tree based on the PCG123. By the way, the result of PCGAA was a significant difference from the previous study and taxonomic system (Figure. 4). So, we discard it. The monophyly of Iridopterygidae and Gonypetidae were strongly supported again. By comparing the values of the clades, we find the value of the clades in the phylogenetic relationship based on the PCG123 dataset was higher than the other. Hence, we select it as the final result.

Secondly, via DAMBE 7 [4], we found a slight saturation of third codon positions (table 1)

Table 1. Saturation testing in phylogenetic analyses.

Gene rigions

NumOTU

Iss

Iss.cSym

P

Iss.cAsym

P

1st condons

32

0.209

0.756

0.0000

0.548

0.0000

2nd condons

32

0.108

0.756

0.0000

0.548

0.0000

3rd condons

32

0.529

0.756

0.0000

0.548

0.0000

All condons

32

0.290

0.781

0.0000

0.573

0.0000

Thirdly, according to previous research [5-12], Although there was some saturation of the third codon, these research used PCG12, PCG123, etc to discuss the phylogenetic relationship within Mantodea. Furthermore, the result of PCG123 was practically consistent with the result of PCG12 or PCG123R (13 PCGs including all codon positions and RNA and rRNA). In addition, the value of most clades in the result of PCG123 was higher than the result of PCG12 in Zhang et al [6]. Hence, we think it is fine to use the PCG123 dataset to discuss the phylogenetic relationship within Mantodea in this study. Meanwhile, this study focuses on the structure of mitogenomes of Mantodea.

Subset

Nucleotide sequence alignments

Subset partitions

Best model

Partition 1

ATP6_gb_codon1, NAD6_gb_codon1, NAD3_gb_codon1

GTR+I+G

Partition 2

CYTB_gb_codon2, COX3_gb_codon2, COX2_gb_codon2, ATP6_gb_codon2, NAD3_gb_codon2, NAD6_gb_codon2, NAD2_gb_codon2

GTR+I+G

Partition 3

NAD6_gb_codon3, ATP6_gb_codon3, ATP8_gb_codon3, NAD3_gb_codon3

TIM+G

Partition 4

ATP8_gb_codon1, ATP8_gb_codon2, NAD2_gb_codon1

HKY+I+G

Partition 5

COX2_gb_codon1, COX1_gb_codon1

GTR+I+G

Partition 6

COX1_gb_codon2

TVM+I+G

Partition 7

COX3_gb_codon3, CYTB_gb_codon3, COX1_gb_codon3, COX2_gb_codon3

GTR+I+G

Partition 8

CYTB_gb_codon1, COX3_gb_codon1

GTR+I+G

Partition 9

NAD1_gb_codon1

GTR+I+G

Partition 10

NAD4L_gb_codon2, NAD1_gb_codon2, NAD5_gb_codon2, NAD4_gb_codon2

GTR+I+G

Partition 11

NAD4L_gb_codon3, NAD1_gb_codon3

TRN+G

Partition 12

NAD2_gb_codon3

HKY+G

Partition 13

NAD4L_gb_codon1, NAD5_gb_codon1, NAD4_gb_codon1

GTR+I+G

Partition 14

NAD4_gb_codon3, NAD5_gb_codon3

TVN+G

Furthermore, we used IQTREE 2 and MrBayes 3.2 to construct the phylogenetic relationship of Mantodea based on the PCG123R dataset including all codon positions and two rRNAs (Figure. 6,7). The monophyly of Iridopterygidae and Gonypetidae were strongly supported again. Basically, the main topology is consistent with the BI tree and ML tree based on the PCG123. Definitely, the value of these clades based on the PCG123R dataset was slightly higher than the previous result. However, in this study, the phylogeny within Mantodea is not the most important point. Thank u for reading.

  1. author should update to PartitionFinder 2. the version 1 is older.

Response:Thank you for your comments. We revised the content of the article according to the suggestion. The results obtained are consistent with the previous one.

  1. different substitution model and partition might obtain different phylogenetic relationships, the author use gene prior to decide partition scheme. How about codon partition? because we know the third position often highly saturation

Table 2. The partition schemes and best-fitting models selected of 13 protein-coding genes

Response:Thank you for your comments. the codon partition has been displayed in table2. We simultaneously performed gene and codon partitions. Thank you very much.

Although there was some saturation of the third codon, these research used PCG12, PCG123, etc to discuss the phylogenetic relationship within Mantodea. Furthermore, the result of PCG123 was practically consistent with the result of PCG12 or PCG123R (13 PCGs including all codon positions and RNA and rRNA). In addition, the value of most clades in the result of PCG123 was higher than the result of PCG12 in Zhang et al [6]. Hence, we think it is fine to use the PCG123 dataset to discuss the phylogenetic relationship within Mantodea in this study. Meanwhile, this study focuses on the structure of mitogenomes of Mantodea.

Results section

Gene rearrangements
1.lines 185-198, please indicate the symbols meaning of first "*" or "'" in the text, I do not like to guess what they are.

Response:Thank you for your comments. We revised the content of the article according to the suggestion.

2.Figure 2. how decide which one is original transfer gene, and another one is pseudogene?

Response:Thank you for your comments. the length and anticodon of pseudogenes is usually different from the normal one. Via MEGA 7.0, the related genes were further confirmed by alignment to the orthologous gene regions of other praying mantis mitogenomes. Hence, we can confirm which is the pseudogene.

  1. Figure 3. about the box D, the case Theopoma milligratulata show the structure between trnM to trnI. This is not correct region in CR.

Response:Thank you for your comments. In this region, the gene arrangement of Theopoma milligratulata was “M-CR-I-Q-M*”. So, it is right.

  1. the resolution is provided figures is not enough, and the word in the figure could not be searched. This is helpful to readers to search certain taxa.

Response:Thank you for your comments. We will provide the editorial team with better images.  It is the journal term work, I think they will help us with this.

  1. the scientific name could not be searched in Figure 4, this is hard to find certain taxa if readers are interested in them.

Response:Thank you for your comments. We will provide the editorial team with better images.  We think they will help us with this.

Discussion
1. lines 284-286, the explaination of TDRL should be place in "introduction" section. This work is address on this part, but it is weird that the explaination is showed in discussion section

Response:Thank you for your comments. We revised the content of the article according to the suggestion.

Some reference in this response.

  1. Minh, B.Q.; Schmidt, H.A.; Chernomor, O.; Schrempf, D.; Woodhams, M.D.; Von Haeseler, A.; Lanfear, R. IQ-TREE 2: new models and efficient methods for phylogenetic inference in the genomic era. Mol. Biol. Evol. 2020, 37, 1530-1534.
  2. Ronquist, F.; Teslenko, M.; Van Der Mark, P.; Ayres, D.L.; Darling, A.; Höhna, S.; Larget, B.; Liu, L.; Suchard, M.A.; Huelsenbeck, J.P. MrBayes 3.2: efficient Bayesian phylogenetic inference and model choice across a large model space. Syst. Biol. 2012, 61, 539-542.
  3. Stamatakis, A. RAxML version 8: a tool for phylogenetic analysis and post-analysis of large phylogenies. Bioinformatics 2014, 30, 1312-1313.
  4. Xia, X.H. DAMBE7: New and Improved Tools for Data Analysis in Molecular Biology and Evolution. Mol. Biol. Evol. 2018, 35, 1550-1552.
  5. Xu, X.D.; Guan, J.Y.; Zhang, Z.Y.; Cao, Y.R.; Storey, K.B.; Yu, D.N.; Zhang, J.Y. Novel tRNA gene rearrangements in the mitochondrial genomes of praying mantises (Mantodea: Mantidae): Translocation, duplication and pseudogenization. Int. J. Biol. Macromol. 2021, 185, 403-411.
  6. Zhang, L.P.; Yu, D.N.; Storey, K.B.; Cheng, H.Y.; Zhang, J.Y. Higher tRNA gene duplication in mitogenomes of praying mantises (Dictyoptera, Mantodea) and the phylogeny within Mantodea. Int. J. Biol. Macromol. 2018, 111, 787-795.
  7. Zhang, L.P.; Yu, D.N.; Cheng, H.Y.; Zhang, J.Y. Data for praying mantis mitochondrial genomes and phylogenetic constructions within Mantodea. Data Brief 2018, 21, 1277-1285.
  8. Zhang, L.P.; Cai, Y.Y.; Yu, D.N.; Storey, K.B.; Zhang, J.Y. Gene characteristics of the complete mitochondrial genomes of Paratoxodera polyacantha and Toxodera hauseri (Mantodea: Toxoderidae). PeerJ 2018, 6, e4595.
  9. Wang, J.-J.; Yang, M.-F.; Dai, R.-H.; Li, H.; Wang, X.-Y. Characterization and phylogenetic implications of the complete mitochondrial genome of Idiocerinae (Hemiptera: Cicadellidae). Int. J. Biol. Macromol. 2018, 120, 2366-2372.
  10. He, B.; Su, T.; Wu, Y.; Xu, J.; Huang, D. Phylogenetic analysis of the mitochondrial genomes in bees (Hymenoptera: Apoidea: Anthophila). PLoS ONE 2018, 13, e0202187.
  11. Bai, J.; Xu, S.; Nie, Z.; Wang, Y.; Zhu, C.; Wang, Y.; Min, W.; Cai, Y.; Zou, J.; Zhou, X. The complete mitochondrial genome of Huananpotamon lichuanense (Decapoda: Brachyura) with phylogenetic implications for freshwater crabs. Gene 2018, 646, 217-226.
  12. Dias, C.; Lima, K.d.A.; Araripe, J.; Aleixo, A.; Vallinoto, M.; Sampaio, I.; Schneider, H.; Rêgo, P.S.d. Mitochondrial introgression obscures phylogenetic relationships among manakins of the genus Lepidothrix (Aves: Pipridae). Mol. Phylogenet. Evol. 2018, 126, 314-320.
